# Noise Levels and Sleep in a Surgical ICU

**DOI:** 10.3390/jcm11092328

**Published:** 2022-04-22

**Authors:** Maria Guisasola-Rabes, Berta Solà-Enriquez, Andrés M. Vélez-Pereira, Miriam de Nadal

**Affiliations:** 1Anaesthesiology Department, Hospital Vall d’Hebron, Universitat Autònoma de Barcelona, Passeig de la Vall d’Hebrón 119-129, 08035 Barcelona, Spain; bsola@vhebron.net (B.S.-E.); minadal@vhebron.net (M.d.N.); 2Departamento de Ingeniería Mecánica, Facultad de Ingeniería, Universidad de Tarapacá, Avenue 18 de Septiembre 2222, Arica 1000007, Chile; avelezp@academicos.uta.cl

**Keywords:** sleep, noise, perioperative care

## Abstract

Sleep is disturbed in critically ill patients and is a frequently overlooked complication. The aim of our study is to evaluate the impact of sound levels in our surgical ICU on our patients’ sleep on the first night of admission. The study was performed in a tertiary care university hospital, in a 12-bed surgical ICU. Over a 6-week period, a total of 148 adult, non-intubated and non-sedated patients completed the study. During this six-week period, sound levels were continuously measured using a type II sound level meter. Sleep quality was evaluated using the Richards–Campbell Sleep Questionnaire (RCSQ), which was completed both by patients and nurses on the first morning after admission. A non-significant correlation was found between night sound levels and sleep quality in the overall sample (r = −1.83, 95% CI; −4.54 to 0.88, *p* = 0.19). After multivariable analysis, a correlation was found between higher sound levels at night and lower RCSQ evaluations (r = −3.92, 95% CI; −7.57 to −0.27, *p* = 0.04). We found a significant correlation between lower sound levels at night and a better quality of sleep in our patients; for each 1 dBA increase in LAFeq sound levels at night, patients scored 3.92 points lower on the sleep questionnaire.

## 1. Introduction

We know that sleep is disturbed in critically ill patients, both in quantity and quality [1,2,3,4,5]. The cause for these sleep disturbances in the ICU are thought to be multifactorial, with environmental factors probably being the most important. The environmental factor most often cited in the literature as a sleep disturbing factor is noise [2,6,7,8]. Other contributing factors include but are not limited to: the administration of sleep altering medications, how the intensive care unit (ICU) environment is structured and the effects of acute illness [9]. Sleep, sometimes an undervalued element, is in fact a physiological human need that plays a vital role in maintaining good health and is necessary for human survival.

A frequently overlooked complication in ICU patients is the lack of adequate sleep. Over 50% of these patients may experience some form of sleep disturbance [10], consisting of reductions in several sleep stages, marked sleep fragmentation, circadian rhythm disorganization, and daytime sleepiness [11]. Several studies have demonstrated that poor sleep quality can lead to physical and psychological symptoms [12,13,14]. Although many consequences of sleep deprivation are probably unknown, we know that a lack of sleep elicits adverse effects on the cardiovascular, metabolic and respiratory systems, as well as immune function disturbances and neuropsychological impairment, leading to poor cooperation and cognitive dysfunction [15]. These negative effects can hinder our efforts in treating critical patients, as they make weaning from assisted ventilation difficult, delay patient recovery and ultimately decrease the chances for a positive outcome [16,17,18].

Environmental noise has often been addressed as a relevant contributing factor to the alteration of sleep in the ICU, but the relative contribution of noise to sleep disturbance is often debated when compared to other disrupting factors. In fact, a recent systematic review concluded that it is currently not possible to be able to correctly quantify the extent to which noise is a contributor to sleep disruption in ICU patients [19]. The most frequent sources of sound disruption have been reported as being due to staff conversations, alarms and those related to patient care interventions [20,21].

According to the guidelines from the World Health Organization (WHO), sound levels should not rise above 35 A-weighted decibels (dBA) in hospital areas where patients are treated or observed, allowing for a maximum night level below 40 dBA in order to ensure proper sleep. Several ICU studies have reported sound levels with mean values of 53–59 dBA, with peak sound levels of up to 67–86 dBA [2,21]. ICU noise has been demonstrated to contribute to patients’ lack of REM sleep [2].

Quality of sleep can be evaluated using a variety of means, including both objective and subjective techniques. Within the objective techniques, the gold standard for the measurement of sleep is polysomnography (PSG).

When compared to other measures of sleep, the use of subjective methods for sleep assessment, such as sleep questionnaires, which can be filled in by patients or nurses, is simple, inexpensive and practical. The Richards–Campbell Sleep Questionnaire (RCSQ) was developed for this purpose. The RCSQ is a five-item visual analogue scale that is widely used to evaluate the quality of sleep among intensive care patients. This scale has been tested and found reliable and valid in various studies, including ICU patient populations [22,23,24]. The questionnaire can be filled in by both patients or nurses.

Many studies have used interventions to improve patients’ quality of sleep which usually comprises a mix of interventions, including reducing the levels of both light and noise or listening to soft music. The aim of our study was to assess the impact of sound levels in our ICU on our patients’ sleep on their first night of admission. We also looked at the difference between the patient’s and nurse’s perception of quality of sleep.

## 2. Materials and Methods

After obtaining local ethical approval (PR(AG)258/2017), we conducted a prospective single-centre study. The study took place in the surgical ICU of a tertiary university hospital, which consisted of a 12-bed unit, during a six week period [25]. Our surgical ICU operated in three shifts: a morning shift (8.00–15.00), an afternoon shift (15.00–22.00), and a night shift (22.00–8.00). During the period between the 8th of November and the 21st of December of 2017, 154 consecutive patients were included in the study. Exclusion criteria were as follows: patients under mechanical ventilation and/or sedation requirement at ICU admission, those undergoing cranial neurosurgery or with a Glasgow Coma Scale score less than 14 and patients who were unable to read or understand a sleep questionnaire. Patients who had already been enrolled in the study and returned to ICU for another reason were also excluded.

The study had an analytical and observational design. Over the six-week period, sound levels were continuously measured using a type II sound level meter (SC420, CESVA Instruments, Barcelona, Spain). We recorded the following data every 1 s with the filter frequency in A-weighting and Fast mode: noise equivalent level (LAFeq) to establish the noise level, the maximum level (LAFmax) in order to establish the maximum value, and the minimum level (LAFmin), which is used to analyse temporary variations in noise in the unit. Additionally, the 90th percentile (LAF90) and the peak noise levels (LApeak) were also recorded in order to establish the background noise levels and the dynamic of peak times or random noises, respectively [26,27]. The sound level meter was positioned following the recommendations of Fortes-Garrido [28], in an attempt to achieve the sound level samples that would be the most representative of the sound levels in our surgical ICU. We placed the sound level meter in the centre of the surgical unit, 55 cm from the ceiling and 110 cm from the wall. We also took into account the internal ICU dynamics, in order to try to interfere as little as possible with daily clinical activity [25].

Sleep quality was evaluated using the Richards–Campbell Sleep Questionnaire (RCSQ) [29], which was completed by the patients and the assigned nurse. Patients were asked to mark, with a pen, on a scale of 0–100 mm, their perception of quality of sleep in response to 5 questions (Appendix A). In this validated assessment instrument, higher scores indicate better sleep. Over the 6-week study period, on a daily basis, between 7.00 and 8.00 h, patients and nurses completed the questionnaire. To prevent adaptation to noise, we only analysed the RCSQ results from the patients’ first night of ICU stay. In addition, other factors that could influence the patients’ sleep were recorded: age, sex, body mass index (BMI), type of surgery (elective or urgent), hourly visual analogue scale (VAS) for pain intensity and number of hours with a VAS score higher than 3. Other factors that might influence the patients’ sleep, such as nasogastric tube or urinary catheter insertion, overnight postural and/or drain change and hypnotic drug prescription, were also recorded (Table 1).

### Statistical Analysis

We performed a descriptive analysis of the demographic variables. Continuous variables were expressed as mean ± SD, and categorical variables were expressed as absolute values and percentages. Quantitative data were compared with a one-way analysis of variance (ANOVA) test for data showing normal distribution and the Kruskal–Wallis test was used for categorical variables for data that were not normally distributed. Testing for normal distribution of the data was performed by the Kolmogorov–Smirnov test. Noise measurement data (LAFeq, LApeak, LAFmin, LAFmax and LAF90) were aggregated and summarized from seconds to hours and day periods (morning, afternoon and night) and analysed using a linear regression model adjusted by hours and periods. We also performed a subgroup analysis to study sound measurements during the nighttime.

To determine which factors were associated with a poor response on the RCSQ, we conducted an analysis to determine the relationship between sleep (dependent variable) and sound level indicators (independent variable) and used patient characteristics (shown in Table 1) as incidence covariables. The association between RCSQ values and noise levels was tested using Pearson’s correlation, whereas the association between RCSQ values obtained by nurses and patients was analysed using Pearson’s correlation and Bland-Altman plots. The distribution and dispersion of data were examined before inferential analyses were performed.

Data analyses were carried out using the R version 3.4.1 statistical software package (R Foundation for Statistical computing, Vienna, Austria) and a value of *p* < 0.05 was considered to be statistically significant.

## 3. Results

During the study period, 148 patients were included in our study (Table 1); six patients with inclusion criteria were discharged to a hospital ward before they could complete the questionnaire. A total of 60.14% were male and the mean age of all patients was 63 ± 15 years. The mean duration of ICU stay was 26.2 ± 2.3 h. The average nighttime (from 22:00–08:00 h) sound levels in the 6-week study period are shown in Table 2.

The reason for ICU admission was elective surgery in 111 patients (75%), with general surgery being the most frequent type. A total of 45 patients (30.41%) reported a VAS for a pain score higher than 3 during 2 or more hours and 54 patients (36.48%) received hypnotic drugs to aid sleep.

A non-significant correlation was found between night sound levels and sleep quality in the overall sample (r = −1.83, 95% CI; −4.54 to 0.88, *p* = 0.19). The correlation between RCSQ scores and sound levels at night depending on the covariables studied are shown in Figure 1. When conducting the multivariable analysis, we only found two variables that influenced patients’ quality of sleep. Patients with a VAS score > 3 (45 patients), that is, those that presented with VAS scores ranging from 4–10 during 2 or more hours, and those receiving hypnotic drugs (54 patients) showed the lowest RCSQ values (*p* = 0.05 and *p* < 0.01, respectively). When patients with VAS scores > 3 and those receiving hypnotic drugs were excluded from the analysis, a correlation was found between higher sound levels at night and lower RCSQ scores (r = −3.92, 95% CI; −7.57 to −0.27, *p* = 0.04) (Figure 2a).

Regarding the nurses’ and patients’ perception of the patients’ sleep, a correlation was found between the two RCSQ evaluations (r = 0.43 (0.29–0.55), *p* < 0.001) (Figure 2b). Despite this agreement, nurses tended to overestimate their patients’ sleep by more than 8 points on the RCSQ score (Figure 2c). Nurse’s RCSQ scores were used to compare them with patient’s RCSQ scores and not as an independent measurement; thus, only a comparative analysis was performed between both results.

## 4. Discussion

The acoustic status of our ICU, indicated by LAFeq values, was demonstrated to be similar to the situation described in other studies [30]. In our study, the results demonstrated that once potentially confounding factors such as pain had been excluded, lower sound levels led to better sleep quality, indicated by higher RCSQ scores.

Although it seems clear that noise has a negative effect on the quality of sleep of patients in the ICU, more studies are needed to quantify the effect of noise on sleep. Various studies include noise as a factor for sleep disturbance but say it is not the predominant factor, citing other factors such as pain or the inability to lie comfortably as major contributors [2,31], while other studies defend noise as the main factor for sleep disruption [1,32,33].

An important problem in ICU environments is that certain noises, such as those from ventilator or monitor alarms, as recorded by LApeak (the measure of the maximum instantaneous sound pressure value reached), are probably non-modifiable or very difficult to modify in an ICU environment. Aaron et al. demonstrated a significant correlation of sound peaks of over 80 dBA with arousals from patients’ sleep [34]. At the same time, background noise (as measured by LAF90), has been demonstrated to also be negatively associated with sleep and is often dependent on the building architecture; a recent multicentre study pointed out the importance of taking building properties into account when designing an ICU [26].

No interventions took place in our study to better aid sleep in our patients, but several studies have looked at interventions and mechanisms to promote sleep, with some demonstrating improvements in the quality of sleep [30,32,35,36,37,38]. The most common include the use of earplugs and eye masks, strategies for noise and light level reductions or the use of relaxing music [32,37,38,39,40]. Non-pharmacological interventions carried out by nurses could also prove useful in helping aid sleep, such as modifying the time of certain patient-care related activities [39].

It became apparent in our study that patients scoring a VAS ≥ 3 for a prolonged period and those receiving hypnotic drugs scored significantly lower. When these patients were excluded, a significant correlation was found between higher sound levels at night and poorer RCSQ scores the following morning. For each 1 dBA increase in LAFeq, patients’ RCSQ scores were almost 4 points lower. It seems evident that pain is an important factor in sleep disruption, regardless of how noisy the ICU environment is, but patients who received hypnotic drugs (usually benzodiazepines) to aid sleep also scored worse on the sleep questionnaire. This may be because these patients already presented with sleep difficulties and had been prescribed hypnotic drugs at home, the reason for which these drugs were prescribed in our unit on admission. A patient with sleep difficulties at home is likely to score worse on a sleep questionnaire than those who do not have difficulty sleeping. An indication for the use of hypnotic drugs may be precisely that the patient demonstrates particular difficulty sleeping on the night of admission and requires medication to aid sleep, which could also be a reason for why these patients scored worse on the sleep questionnaire. One of our major limitations was that we did not identify the patients that already presented sleeping difficulties at home and we also did not account for patients presenting with obstructive sleep apnea syndrome (OSAS) or with pre-existing sleeping disorders. Age, sex, obesity, type of surgery and other factors that could influence a patient’s sleep, such as nasogastric tube or urinary catheter insertion and overnight postural or drain change, did not correlate with a poorer RCSQ score. However, disease severity, as indicated by the APACHE score, was not taken into account when patients were selected, making this a limitation, as there is a possibility that beyond pain, disease severity may also disrupt sleep in ICU patients. Another limitation was that the sound level meter was not placed by the patient bedside; instead, it was placed in the centre of the ICU, very close to the nurses’ station. This allows for closer recordings of staff activities and conversations rather than ventilator and monitor-related noise. Finally, we did not analyse sleepiness during the daytime, but only nighttime quality of sleep. Circadian rhythms in ICU patients are probably altered and daytime sleep should be taken into consideration. We also could have included a questionnaire to be filled in by patients about the perceived causes of sleep disruption during their ICU stay and could have looked at other contributing factors for sleep disturbance.

We also asked the nursing staff to express their perception of how the patient had slept the previous night by completing the RCSQ themselves, and the RCSQ demonstrated a correlation between the patients’ and nurses’ assessments. However, nurses tended to overestimate the quality of sleep. Therefore, it could be of value to inform the nursing staff about the results and impact of noise on the sleep quality of their patients so they become implicated in promoting sleep.

An alternative to polysomnography, the gold standard for sleep measurement, is actigraphy, a validated method for measuring total sleep time as well as sleep fragmentation [41]. However, when comparing actigraphy to nurse assessment, patient questionnaires and PSG, it tended to overestimate total sleep time and sleep efficiency [42] and awakenings were less frequently reported. Bispectral index (BIS), which is an EEG-derived method used to assess the depth of sedation, mainly used during general anaesthesia for patients undergoing surgery, has been evaluated as an alternative to assess the quality of sleep. A recent study demonstrated that BIS monitors could be useful in providing a measure of the depth of sleep, particularly in situations such as intensive care units, and that it could serve as an alternative method for monitoring sleep [43]. The use of PSG in the ICU presents multiple challenges, such as the need for a skilled person to perform, interpret, and score the results. In addition, the use of PSG is expensive and inconvenient [39].

Although the RCSQ is a validated and reliable scale to assess patients’ sleep, future studies should consider using more objective methods, such as the Bispectral Index (BIS), which are widely available, easy to interpret and could potentially prove to be an alternative to PSG.

Once confounding factors were excluded, we found a significant correlation between lower sound levels at night and a better quality of sleep in our patients; for each 1 dBA increase in LAFeq noise levels at night, patients scored 3.92 points lower on the sleep questionnaire.

## Figures and Tables

**Figure 1 jcm-11-02328-f001:**
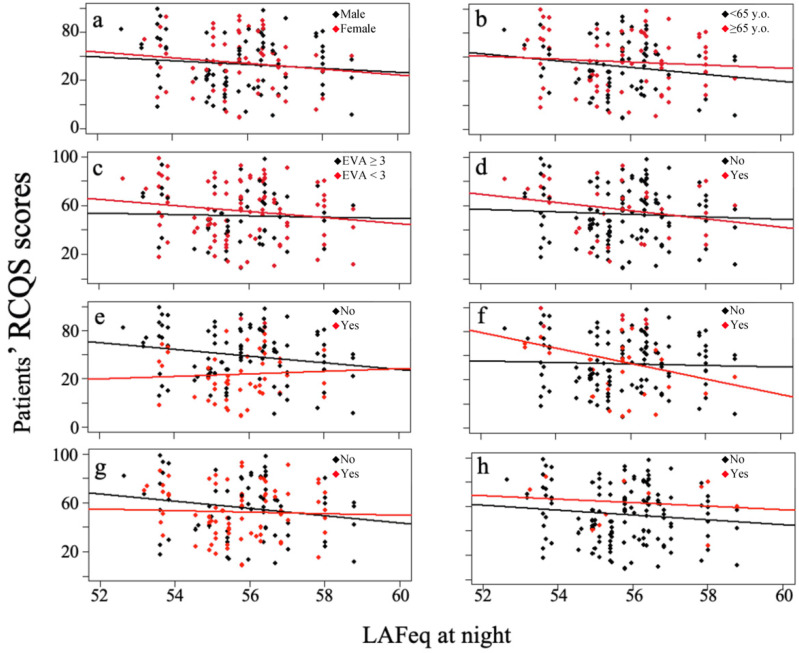
Correlation between patient RCSQ scores and LAFeq at nighttime depending on the covariables; (**a**) gender, (**b**) age, (**c**) VAS ≥ 3 during 2 or more hours, (**d**) urgent surgery, (**e**) hypnotic drug administration, (**f**) nasogastric tube, (**g**) water intake and (**h**) family visit.

**Figure 2 jcm-11-02328-f002:**
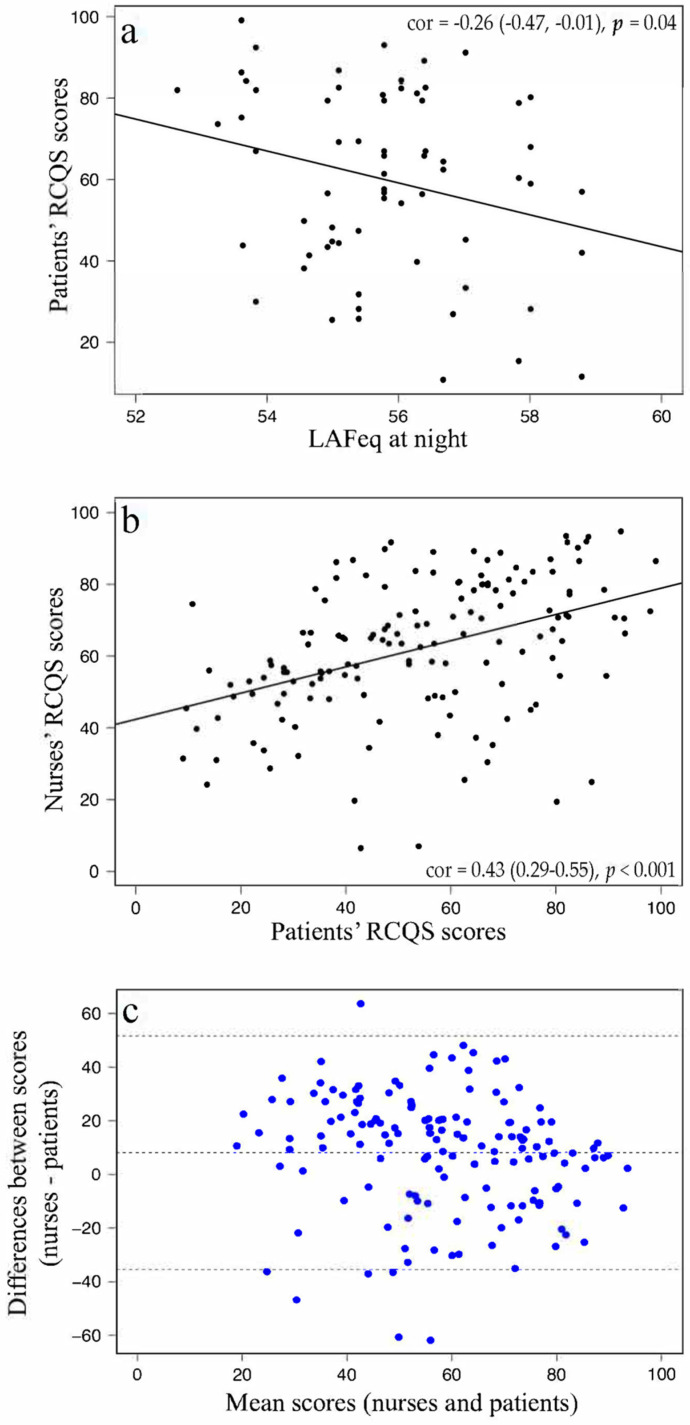
(**a**) RCSQ and mean sound levels at night; (**b**,**c**) nurses’ perception of patients’ sleep compared to the patients’ perception.

**Table 1 jcm-11-02328-t001:** Patient characteristics on inclusion.

Characteristic	
Patient characteristics	Patient included (*n* = 148)
Male sex (*n*; %)	89; 60.1%
Age (mean ± SD)	63 ± 15 year
Weight (mean ± SD)	73 ± 16.8 kg
Height (mean ± SD)	170 ± 8.9 cm
Type of surgery	
Abdominal (%)	44.59
Urologic (%)	14.86
Vascular: extremities/carotid (%)	13.51
Other (%)	10.81
Thoracic (%)	9.46
Spinal (%)	5.41
ENT or maxillofacial (%)	1.35
Surgery characteristics	
Urgent surgery, yes (%)	23.7
Nasogastric tube, yes (%)	22.3
Urinary catheter, yes (%)	88.6
Medical care	
Drain changes, yes (%)	39.2
Hypnotic drug administration, yes (%)	36.5
Initiate drinking water, yes (%)	53.4
Night hygiene, yes (%)	71.0
Postural changes, yes (%)	33.4
VAS pain score	
Maximum (mean ± SD)	(2.5 ± 2.8)
Minimum (mean ± SD)	(0.2 ± 0.6)
No. of hours VAS was >3 (mean ± SD)	(0.9 ± 1.5)

VAS: Visual analogue scale. ENT: ear, nose and throat.

**Table 2 jcm-11-02328-t002:** Sound levels during the 6-week period.

	LAFeq	LAFpeak	LAFmin	LAF90	LAFmax
Mean	57.3 dB	75.6 dB	52.1 dB	52.7 dB	61.6 dB
Median	58.3 dB	76.6 dB	52.7 dB	53.3 dB	62.7 dB

## Data Availability

Not applicable.

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
