# Peer review of "Noise Levels and Sleep in a Surgical ICU"

_jcm, 2022, doi:10.3390/jcm11092328_

Round 1
Reviewer 1 Report
I congratulate for this interesting research in ICU. Nevertheless, the reference 20 is not so recent. I suggest to add the reference Elbaz et al. Annals of Intensive care Article number 25 (2017).
The lack of the analysis of sleepiness during the day is a problem. You must show us some results in daytime to compare if you found a phenotype of patients in this population patient's.
How did you analyse the sound for example of the mechanical ventilation? Did you listen it?
Reviewer 2 Report
I had the pleasure to review the manuscript entitled “Noise levels and sleep in a surgical ICU”. This is a prospective single-center study. The main objective of this study was to assess the impact of sound levels on ICU patients’ sleep on their first night of admission. One hundred and eight ICU patients who underwent surgeries for a variety of indications and were neurologically stable to read and understand the questionnaire were included in the analyses.
I think the findings are important although topic is not novel, but substantial revision is required before publication. Please see my comments below:
- Introduction is too long. I was lost reading it. I recommend substantially shortening it. Paragraph about PSG can be deleted or shortened and/or moved to Discussion.
- It is not clear whether the same analyses were performed for nurse’s RCSQ evaluations. If not, it would be interesting to see the results.
- Page 5, figure 1 and its legend is confusing, please clarify what readers should understand from these graphs.
- The main limitation of this study is that the existence of prior sleep problems is not taken into consideration as pointed out by authors. Without excluding this important factor, it is hard to link sleep problems to ICU noise. It is noteworthy that ICU noise was correlated with sleep quality after eliminating pain and hypnotic drug intake. However, hypnotic drug use was frequent (36.48%). I do not think that this merely represents prior use. Please describe indications for use of hypnotic drugs in these patients in more detail.
- Another confounding factor could be the disease severity in these patients. If available, APACHE scores to categorize disease severities of patients would be more appropriate to compare since there is a possibility that beyond pain, disease severity also may disrupt sleep in ICU patients.
- A brief questionnaire filled out by the patients about the causes of sleep disturbances during ICU stay could also be helpful to identify the perceived effect of noise and other factors. This can be stated as a limitation if not performed already.
- Contrary to Introduction, Discussion is short. I recommend adding a few more paragraphs for in-depth discussion of the relation between sleep quality and ICU noise levels. Also please discuss possible mechanisms and management alternatives to overcome sleep problems in the ICU. There are many papers in the literature (for example, some recent papers can be cited: DOI: 10.1016/j.wneu.2020.05.017DOI: 10.1111/jan.14914 DOI: 10.1097/CNQ.0000000000000240)
Reviewer 3 Report
It would be interesting to know if patients with sleep disordered breathing (OSAS) were included in the study. did the authors correct for pre-existing sleep disorders. if not this should also be mentioned in the discussion.
148 patients were included, but in the methods section it is written tthat 154 patients were included. which one is true?
Round 2
Reviewer 1 Report
Dear authors,
I congratulate the authors for this very interesting study in the field of ICU and Sleep.